# Investigations of Histomonosis-Favouring Conditions: A Hypotheses-Generating Case-Series-Study

**DOI:** 10.3390/ani13091472

**Published:** 2023-04-26

**Authors:** Julia Lüning, Amely Campe, Silke Rautenschlein

**Affiliations:** 1Clinic for Poultry, University of Veterinary Medicine Hannover, Foundation, Buenteweg 17, 30559 Hannover, Germany; 2Department of Biometry, Epidemiology and Information Processing (IBEI), WHO Collaborating Centre for Research and Training for Health at the Human-Animal-Environment Interface, University of Veterinary Medicine Hannover, Foundation, Buenteweg 2, 30559 Hannover, Germany; amely.campe@tiho-hannover.de

**Keywords:** *Histomonas meleagridis*, histomonosis, turkey, epidemiology, field study, risk awareness

## Abstract

**Simple Summary:**

Histomonosis has become a serious disease not only for turkeys but for the entire poultry sector. Despite a focus on research on histomonosis over recent years, possible disease-favouring conditions are still poorly understood. For this reason, an interview-based case series study was initiated on 31 affected turkey farms in Germany. Despite a generally good risk awareness among participating farmers for disease-favouring conditions, an insufficient attitude towards certain biosecurity aspects was detected. Furthermore, hypotheses concerning various possibilities of pathogen introduction via animate or inanimate vectors could be generated, which need to be tested in follow-up studies. Overall, this study demonstrates that improvements in biosecurity, flock, and health management are necessary to ensure animal welfare in the future.

**Abstract:**

Since the ban of effective feed additives and therapeutics, histomonosis has become an important disease and, subsequently, a welfare issue for turkey production. We conducted an interview-based case series study to generate hypotheses about possible disease-favouring conditions in 31 *H. meleagridis*-infected flocks. The determined parameters were related to the general farm (flock management, biosecurity measures, etc.) as well as the histomonosis-specific disease management. Some inadequate biosecurity measures were observed. An inappropriate usage of the hygiene lock and cleaning as well as the disinfection frequency of equipment, clothes, and the hygiene lock could possibly be histomonosis-favouring conditions. These factors could increase the risk for the introduction of *H. meleagridis* and the risk of a pathogen spread on an affected farm. Insects, wild birds, litter materials, and contaminated dung could be potential vectors of *H. meleagridis*. Predisposing gastrointestinal diseases were observed in 71% of the affected flocks. Additionally, stress events related to higher temperature, movement of birds, and vaccination were documented in association with clinical histomonosis. The results emphasise the need for both good disease control and health management to ensure sustainable animal health and welfare.

## 1. Introduction

*Histomonas meleagridis* (*H. meleagridis*) is the causative agent of histomonosis, also known as blackhead disease. Histomonosis has become an economically important disease throughout the world for poultry, especially for turkeys and chickens, since the ban of effective feed additives and therapeutics in many countries [1,2]. The pathogen may invade the hosts directly or indirectly by oral or cloacal infection [3]. Vectors, such as the caecal worm *Heterakis gallinarum* and earthworms, but also mechanical or inanimate vectors, play an important role in the epidemiology of histomonosis [3,4,5]. Wild birds and numerous poultry species can constitute a reservoir for the pathogen [3,5]. *H. meleagridis* causes pathognomonic lesions primarily in the caeca. Inflammatory reactions and the subsequent damage of the intestinal barrier may lead to a spread of the pathogen to the liver and, less frequently, to other organs [3,6]. The clinical signs are primarily non-specific, but a typical sulfur coloured diarrhoea may be observed [7,8]. The clinical symptoms may appear after an incubation period of 6 to 15 days, starting at the third week of life [3,9]. However, outbreaks of older meat turkeys or breeder turkeys have also been reported [10,11]. Morbidity and mortality rates may range from less than 10% up to almost 100% [3,10,11,12,13,14]. Not all factors contributing to the variable clinical course of histomonosis have been elucidated so far. Pathogen-associated variation in virulence as well as host susceptibility may contribute to the pathogenesis [10,14,15,16,17]. We can speculate that additional factors may also favour the disease development. Only a few epidemiological investigations have been carried out to identify histomonosis-favouring conditions. The seasonality, regionality, age of turkeys, pre-existing gastrointestinal diseases, and litter moisture seem to have a favourable influence on the clinical development of histomonosis [10,15,18]. Due to the difficulty of experimental investigations of possible histomonosis-favouring factors in a multifactorial setting, further field investigations are necessary to understand these factors in affected turkey flocks, as the disease currently causes high economic losses as well as an animal welfare problem worldwide. We implemented a case series study on 31 turkey farms with a history of histomonosis outbreaks to generate hypotheses about histomonosis-favouring conditions. The generated hypotheses could serve as starting points for further studies in different research areas on histomonosis. This case series study was a follow up of a case control study, which investigated possible ways that *H. meleagridis* is introduced on turkey farms with and without cases of histomonosis [19].

## 2. Materials and Methods

### 2.1. Data Collection

A total of 31 out of 40 participating case farms of a previously conducted case control study also contributed to follow-up case series investigations [19]. A questionnaire was designed for an on-farm interview (Appendix A). It was aimed to collect all relevant data on an exemplary histomonosis outbreak since January 2018, which the participating farmers were assumed to remember best. After a testing phase of the designed questionnaire on two farms––which are under the veterinary care of the Clinic for Poultry of the University of Veterinary Medicine Hannover, Foundation, although neither participated in the case control study nor the case series study––the farm visits, including an interview to collect answers for the questionnaire, took place between August 2021 and January 2022. Additionally, a protocol documented all other important information, such as the current conditions in the hygiene lock and the state of health and the management of the housed flock (Appendix A).

### 2.2. Data Analysis

The descriptive analysis was performed using SAS Enterprise Guide, version 7.15, and SAS Software, version 9.4 (SAS Institute Inc., Cary, NC, USA). Graphs were created with GraphPad Prism (Version 6.02 for Windows, GraphPad Software, La Jolla, CA, USA, www.graphpad.com (accessed on 23 October 2022)). The handwritten answers were transcribed, coded, and tabulated in Microsoft Excel (Microsoft Corporation, Redmond, WA, USA, 2018. Microsoft Excel, version 2303, https://office.microsoft.com/excel (accessed on 1 March 2022)). The coding was performed manually according to an in-house scheme. Missing information was prevented through the direct interview. A total of 164 variables (Appendix A) were analysed (number (*n*) and percentage (%) for qualitative variables; median, standard deviation, variance coefficient, minimum, maximum, 25%- and 75%-quantile for quantitative variables). The classification of variables is as follows: description and management of the farm and flock (Appendix A); general biosecurity measures (Appendix A); health management, incidence and therapy of diseases (Appendix A); outbreak management (Appendix A); and coincidental findings (Appendix A). To reduce social desirability bias, we obtained a personal impression of the conditions on the farm and in the affected houses. In addition, all necessary documents for answering the questionnaire, such as the documentation of production data and veterinary reports, were examined. Subsequently, the correctness of the farmers’ answers to the questionnaire was checked and, in case of discrepancies, the answers were corrected after consultation with the farmers [20,21].

## 3. Results and Discussion

In order to generate hypotheses about histomonosis-favouring conditions, associated questions were selected and evaluated. The results are presented and discussed in the following paragraphs.

### 3.1. Description and Management of the Farm and Flock

A total of 30 meat turkey farms participated in our field study. Two of them were rearing farms, 12 were fattening farms, and 16 were combined rearing and fattening farms. In addition, one turkey breeder farm participated (Appendix A). Our results confirm previous field studies in which both fattening and breeder flocks were affected by *H. meleagridis* [10,11,15]. A total of 18 farms had already experienced several outbreaks of histomonosis, which indicates either favouring conditions for a reintroduction of *H. meleagridis* or a pathogen persistence on the farms, possibly in vectors or due to resistant stages in the environment [14,22,23,24,25,26].

Due to the production structure of meat turkeys in Germany, there are often one or more movements of the birds within a farm or between different farms. A total of 32% of the farms had moved a flock up to ten days before an outbreak of histomonosis was noticed. A movement may not only be associated with an increased risk for a pathogen introduction, but can also be considered as a stress-inducing factor [27]. An upregulation of pro-inflammatory factors and factors associated with immunosuppression can be measured in stressed poultry [28,29]. These changes may favour the development of a more severe clinical disease in turkeys infected with *H. meleagridis*. Moreover, stress could lead to a resurgence of *H. meleagridis* in latently affected birds with an increase in pathogen excretion, resulting in a direct lateral pathogen spread with large numbers of affected turkeys [30,31].

Weather conditions, such as a drop in temperature, major precipitation, or wind were either hardly or not observed within the context of a histomonosis outbreak. Most of the analysed outbreaks (24/31) appeared in the warmer season between June and September (Figure 1), confirming previous field studies [4,10,15,32]. The seasonal distribution of outbreaks may suggest an impact of temperature-related stress. As 12 farmers reported extreme drought (Appendix A), it can be assumed that high outside temperatures were present, which challenged the climate management within the turkey houses [33]. Temperature-related stress may be associated with changes in the gut microbiome, which can influence the blood flow and may lead to hypoxia [34,35,36]. Hypoxia may lead, in turn, to a higher permeability of the gut barrier due to morphological and functional damage, resulting in a systemic distribution of *H. meleagridis* and a subsequent severe course of the disease [36]. Temperature-related stress may also influence the composition of the gut microbiome towards microbes that favour *H. meleagridis* [37].

Most of the affected turkey houses were directly surrounded by green areas as well as trees and bushes, which could provide an ideal environment for vectors and wild birds [25,38]. On the other hand, these surrounding conditions may be suitable for emission control, emphasising the controversial requirements for poultry farms [39].

### 3.2. General Biosecurity Measures

A total of 45% of the farmers observed insects, mainly beetles and flies, in the outbreak houses (Table 1), which are considered as mechanical vectors for *H. meleagridis* [4,40,41]. Insect control was reported by 7/31 farmers, leaving 71% of the affected farms with a higher risk of an introduction of *H. meleagridis*. This risk may even be enhanced by rough concrete floors and floors with cracks, which were reported by 74% of the farmers (Appendix A), allowing possible vectors to survive cleaning and disinfection procedures [42]. These circumstances could possibly have led to a re-occurrence of histomonosis.

A total of 27 farms had at least one hygiene lock, while four farms had two (one at the farm entrance and one in front of each turkey house). The equipment of the hygiene lock and the use of suitable clothes is presented in Figure 2. In most cases, a wash basin with soap was available. Nevertheless, in some cases, we observed that the cleaning and disinfection of hands before entering the turkey house were not performed during our personal visit, suggesting irregular use. In only 42% of the farms, a turkey-house specific overall was worn. In the other cases, the farmers wore clothes that were also worn outside the turkey houses. Furthermore, it was noticeable that one-way products, such as footwear, overalls, gloves, and hairnets, were rarely used. Commendably, footwear only worn in the turkey house was used on 30 farms, for which disinfection basins were available on 26 farms.

In the majority of farms, clothes were cleaned at least weekly (20/31; Appendix A). However, the turkey-house specific footwear was wet-cleaned less frequently than weekly in more than 70% of the farms, and disinfection basins, if present, were also cleaned and refilled less frequently than weekly in 73% of the farms.

A total of 55% of the farms wet-cleaned their hygiene locks at least weekly. However, the hygiene locks were disinfected weekly by only 20% of the farms. On the other hand, regular cleaning (28/31) and disinfection frequency (27/31) of the cadaver storage facilities was observed. This fact is indicative of a good awareness of the participating farmers regarding possible risks associated with cadaver storage, possibly due to advanced education measures in the context of the avian influenza crisis in recent years [27,43].

A total of 30 farms used enrichment materials at the time of an outbreak (Table 2). In addition to materials that are easy to clean (e.g., plastic, metal, rubber), materials that are difficult to clean (e.g., wood, pecking stones) were also used in the majority of cases (approximately 73%). The results of the study indicate that the re-use of materials that are difficult to clean, especially the re-use of pecking stones, on approximately 27% of the farms may represent an avoidable risk for a spread of the pathogen (Appendix A).

The abovementioned analyses of biosecurity measures indicate that there are some deficiencies in the behaviour of farmers with respect to cleaning and disinfection measures that could lead to the introduction of the pathogen *H. meleagridis* or other disease-favouring pathogens into a flock [18,43]. Inadequate hygiene and biosecurity measures could also favour the spread of the pathogen between several houses on a farm. Our study supports this hypothesis, as there were outbreaks in several turkey houses of a farm with an average of 1.5 days in between (range: 0–28 days; Appendix A).

### 3.3. Health Management, Incidence and Therapy of Diseases

The investigated flocks were vaccinated against numerous pathogens (Appendix A). More than 60% of the farms did not continue the general vaccination program during a clinical outbreak of histomonosis. The last vaccination was on average 14 days before the beginning of the respective histomonosis outbreaks (range: 0–78 days; Appendix A). Based on this observation that farmers interrupted the vaccination program in case of clinical histomonosis, it can be concluded that farmers are generally well aware of the risk of the additional stress caused by vaccination. A previous study has already demonstrated that vaccination can be a meaningful stress factor [44].

Pre-existing diseases were diagnosed in 23 flocks (Figure 3). It was noticeable that over 70% of the affected flocks were diagnosed before the onset of a histomonosis outbreak with gastrointestinal diseases, associated with the detection of *Escherichia coli*, *Eimeria* species (spp.), or *Clostridium* spp. (Table 3). Accordingly, almost all flocks were pre-treated with antibiotics (24/31; Appendix A). The abovementioned microbes seem to favour *H. meleagridis*-related clinical symptoms or vice versa [18,45,46,47,48,49]. In addition, the antibiotic treatment could lead to imbalances in the gastrointestinal tract, possibly resulting in an accumulation of microbes that favour *H. meleagridis* [50].

The mortality caused by *H. meleagridis* was on median 25% in the analysed flocks (range: 4–100%; Appendix A). Clinical symptoms occurred on median at 46 days of life (Figure 4) and lasted on median for 21 days (range: 0–98 days; Appendix A). After the onset of clinical symptoms, a pathological examination was immediately initiated on all farms (Figure 4). Numerous other diagnostic tests were performed to identify the causative pathogen *H. meleagridis* and secondary pathogens. It is remarkable that the pathological examinations always led to the diagnosis of histomonosis, possibly due to a good veterinarians’ knowledge about the pathognomonic caecum and liver lesions [7,51]. In addition, the detection of the pathogen DNA was successful in 27/28 cases (Appendix A). Treatment against histomonosis was carried out on 30 farms (Table 4). An antibiotic treatment with Paromomycin (27/31) was used in the majority of cases (Appendix A). Furthermore, herbal products were used in 45% of the farms (14/31). However, only five farmers reported a rapid improvement of flock health, which was combined with a low to medium mortality rate. This corresponds to the results of numerous studies in which both Paromomycin and herbal products had reliable effects in vitro but have not yet shown effectiveness in field outbreaks with severe clinical symptoms [13,52]. In addition, the emergency killing in one third of the cases is also indicative of a low effectiveness of the used prophylactic and therapeutic agents.

Overall, the occurrence of pre-existing diseases and the lack of reliable histomonosis control measures seem to favour the clinical development of histomonosis.

### 3.4. Outbreak Management

All flocks were inspected at least twice a day. Clinically sick flocks were controlled even more frequently in about 65% of the farms (Appendix A). On 29 farms, turkeys with severe clinical symptoms, unable to eat and drink, were killed immediately for animal welfare reasons according to the farmer’s assessment. On almost all farms, the cadavers were removed immediately to cooled and closed containers. Equipment, such as wheelbarrows (12/31) and wheel loaders (5/31), was used in some cases for disposal. The disposal was carried out via different routes, such as through emergency doors (9/31), turkey-house doors (21/31), or the hygiene lock (12/31). Almost all caretakers re-passed the hygiene lock after disposal (28/31). In the event of a histomonosis outbreak, almost all farmers used turkey-house specific clothes for the affected flock. However, this did not include the use of turkey-house specific equipment. Only 61% of the farms used turkey-house specific equipment in the case of a histomonosis outbreak. The use of shared equipment in affected and unaffected houses posed a risk of pathogen transmission [4].

The contaminated dung was spread on surrounding farmland in 6/31 cases. In most cases (23/31), the dung was transported to biogas facilities. Only two farms disinfected the dung before disposal. The presence of multiple bacteria, viruses, fungi, and parasites in the dung and the possibility of an airborne transmission of pathogens from dung-fertilized soil was described in previous studies [53,54,55]. The reported spread of infected dung, which had not been treated to destroy the pathogen, in the immediate vicinity of a turkey house could favour an accumulation of *H. meleagridis* in the environment, and it could have led to the reported re-occurrence of histomonosis on affected farms as well as outbreaks that did not occur in the predisposed season. The open storage of infected dung also poses a risk for pathogen spread and dissemination, because wild birds, searching for feed residue, could take up the contaminated dung.

To conclude, an inadequate handling of contaminated materials may favor the spread of the pathogen in the vicinity of turkey farms, increasing the risk of a histomonosis outbreak.

### 3.5. Coincidental Findings

Although only reported by few farmers, additional coincidental findings should be considered to have contributed the histomonosis outbreaks. Four farmers observed an increased number of earthworms (Appendix A), known to be a vector for *H. meleagridis*, before an outbreak had appeared [56]. At the same time, these four farmers reported waterlogging in the affected turkey houses. The observed waterlogging could have increased the attractiveness of the inside of the turkey house for vectors, as it provides a cooler and more humid environment [57].

On three other farms, wild birds (crows, birds of prey), which can form a reservoir for *H. meleagridis*, were observed inside the turkey house immediately before an outbreak had appeared [38,58,59].

A pathogen introduction may also be possible via contaminated dust particles originating from tillage and harvesting activities in the immediate vicinity, as described for other pathogens [60,61,62,63]. Favourable wind conditions combined with the abovementioned activities were described by five of the interviewed farmers. However, this route of introduction requires a pathogen persistence in the environment in vectors or possibly as a small cyst-like stage, which has only been described in vitro so far [22,23,24,25,64]. In this field study, the distance between turkey farms that were affected at the same time was on median 3.5 km, which could be another indication of a possible airborne transmission (range: 2.0–10.0 km; Appendix A).

Another route of introduction was suspected by five farmers, who used freshly harvested straw as litter material (Appendix A). Numerous pathogens have already been detected in litter material [65,66]. Straw harvested from potentially contaminated fields in the vicinity of an affected farm could in turn be highly contaminated with *H. meleagridis*, whether solitary, adhering to dust particles, or in vectors.

An individual needle vaccination took place on four farms immediately before an outbreak had appeared. The reported vaccination of the flocks may have promoted the clinical onset of histomonosis. Vaccination is always associated with long-term stress, and immunological stress may have led to an imbalance of the microbiome in the caecum and may have disrupted the immune function of the gut, which could favour the clinical disease development of histomonosis [67].

The analyses of coincidental findings demonstrate that further studies are needed to investigate possible modes of pathogen introduction and possible histomonosis-favouring factors in more detail.

## 4. Conclusions

Outbreaks of histomonosis occurred several times as opposed to once on the respective farms, which emphasises the impact of histomonosis on animal health, animal welfare, and economic aspects. The personal visit and comparison of case farms provided important information to generate hypotheses about possible histomonosis-favouring conditions.

The general hygiene measures evaluated were partially insufficient and for this reason can be speculated to favour histomonosis outbreaks in turkeys. Accordingly, we consider that the impact of the hygiene lock and the associated measures of personal cleaning and disinfection should be investigated further. In addition, the cleaning and disinfection frequency of the equipment, the clothes, and the hygiene lock should be carefully evaluated on each farm, and they should be improved to avoid a possible pathogen introduction, if necessary. Our study further suggests that the pathogen introduction through inanimate vectors should be investigated in more detail in follow-up studies.

Moreover, possible histomonosis-favouring, stress-associated factors have to be further analysed, as our study points out that temperature and activities such as movement of birds or vaccination may pave the way for a clinical histomonosis outbreak.

Additionally, gastrointestinal diseases may have a disease-favouring effect on histomonosis, which needs to be investigated further.

The data on mortality rates, insufficient therapy success, and emergency culling clearly indicate that there is an urgent necessity for the identification of *H. meleagridis* control and intervention measures to minimise animal welfare problems related to the disease and the dramatic economic consequences. Therefore, the case series study sets the ground for hypotheses-testing studies.

## Figures and Tables

**Figure 1 animals-13-01472-f001:**
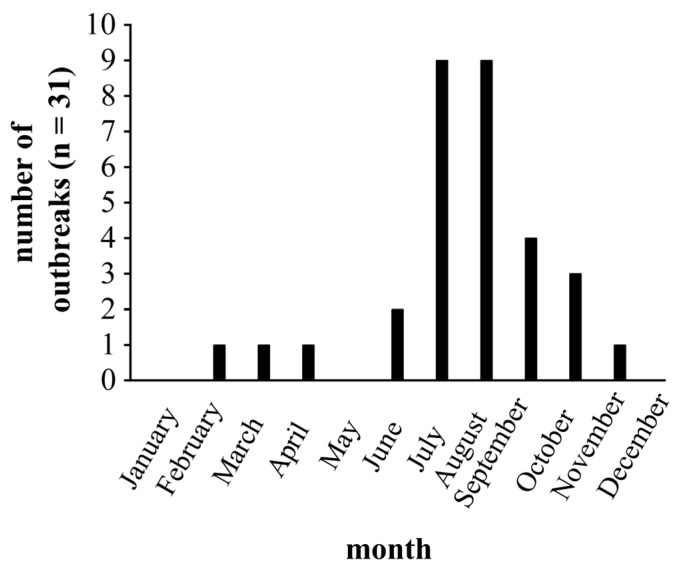
Seasonal distribution of analysed histomonosis outbreaks on 31 farms since January 2018.

**Figure 2 animals-13-01472-f002:**
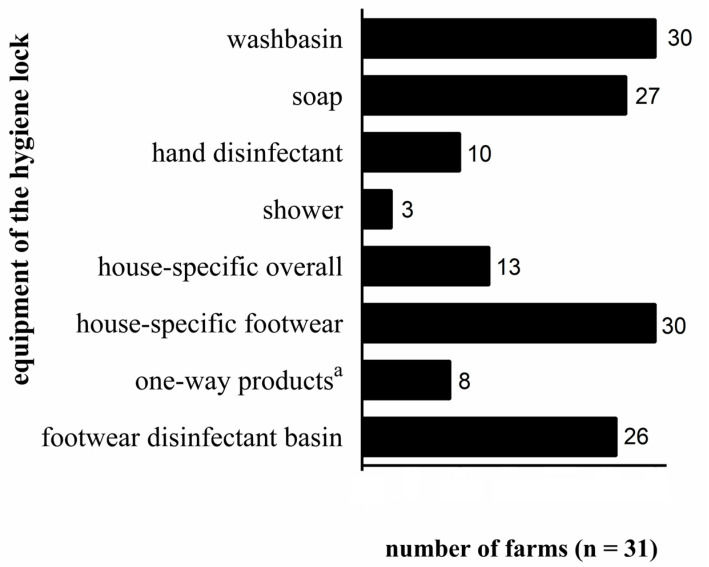
Investigated equipment of the hygiene lock of the analysed farms (*n* = 31) and their availability, ^a^ = one-way gloves and/or -hairnet and/or -footwear and/or -overall.

**Figure 3 animals-13-01472-f003:**
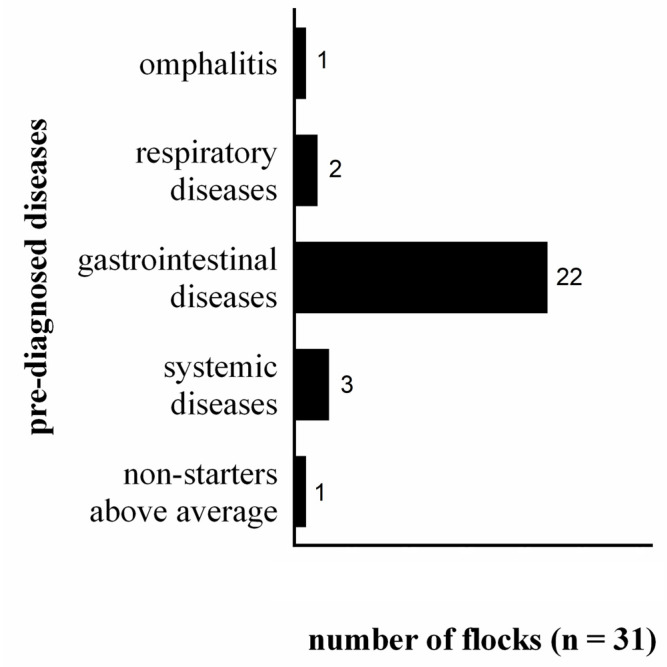
Pre-burden of the analysed flocks (*n* = 31) and their distribution.

**Figure 4 animals-13-01472-f004:**
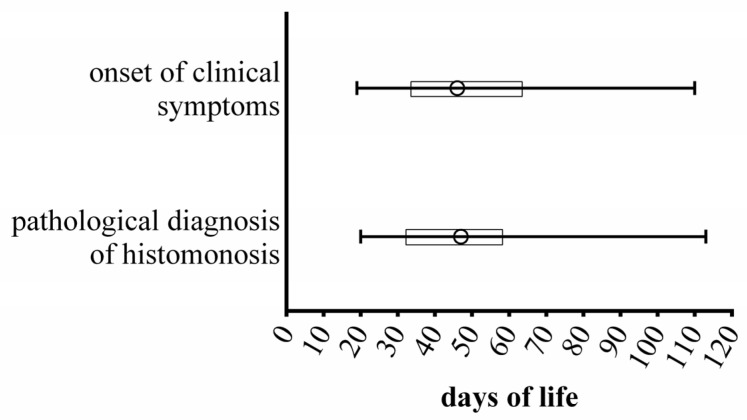
Comparison of the range of onset of clinical symptoms and the range of the pathological diagnosis of histomonosis, o = mean, ☐ = 25%- to 75%-quantile.

**Table 1 animals-13-01472-t001:** Observation and control of insects in the affected turkey houses.

		Observation of Insects		
		Yes	No	Σ
Control of insects	Yes	4	3	7
	No	10	14	24
	Σ	14	17	31

**Table 2 animals-13-01472-t002:** Expected cleaning success of used enrichment materials and the re-use of such materials, *n* = 30 (one farm did not use any enrichment material at the time of the analysed outbreak).

		Enrichment Material		
		Easy to Clean ^a^	Difficult to Clean ^b^	Σ
Re-use of enrichment material	Yes	3	9	12
	No	4	14	18
	Σ	7	23	30

^a^ = plastic, metal, rubber. ^b^ = wood, pecking stones.

**Table 3 animals-13-01472-t003:** Flocks additionally burdened with gastrointestinal pathogens and resulting treatments (occurrence of co-infections with mentioned pathogens was not documented).

	Treatment	
	Yes	No
*Escherichia coli* (*n* = 19)	18 ^a^	1
*Eimeria* spp. (*n* = 14)	14 ^b^	0
*Clostridium* spp. (*n* = 8)	8 ^a^	0

^a^ = treatment with antibiotics. ^b^ = treatment with antiparasitics.

**Table 4 animals-13-01472-t004:** Flock mortality, associated pre-diagnosed gastrointestinal diseases, treatment against *H. meleagridis*, treatment success and need for emergency culling.

Flock Mortality	0–10%	11–50%	51–100%
Number of flocks	8/31	14/31	9/31
Pre-existing gastrointestinal diseases	6/8	9/14	7/9
Treatment against *H. meleagridis* ^a^	8/8	14/14	8/9
Treatment success by rapid clinical improvement	4/8	1/14	0/8
Emergency culling	0/8	5/14	5/9

^a^ = with Paromomycin and/or herbal products and/or analgetics and/or vitamins and/or electrolytes.

## Data Availability

The data presented in this study are available on request from the corresponding author. The data are not publicly available due to privacy restrictions.

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
