# Peer review of "Investigations of Histomonosis-Favouring Conditions: A Hypotheses-Generating Case-Series-Study"

_animals, 2023, doi:10.3390/ani13091472_

Round 1

Reviewer 1 Report

The paper is reporting the results of a field study on Histomonosis in turkeys in Germany. According to the authors, Histomonosis has become an economically important disease for poultry throughout the world, thus th etopic is actual and important for the poultry veterinary field and for farmers.

This is a descriptive study describing the characteristics and conditions in 31 turkey farms in Germany as well as characteristics of the outbreak and outbreak management measures. The data were collected using a structured questionnaire and face to face interviews with farmers during farm visits. Some observations made by interviewers during the farm visits have been also reported.

Descriptions of outbreak herds and disease outbreaks provide valuable information that can be used to design further studies, develop disease control measures or to inform mathematical disease models. Thus the results of present study would be of interest of scientific community and the practitioners of different fields. However the paper needs to be thoroughly re-written before it could be published.

First, the title of the paper is already misleading. As it is a descriptive study - in principle a 'case series study', you can not make any conclusions regarding the effect of the factors you have investigated. In present wording it makes an impression that the aim was to investigate which conditions are favouring histomonis, which is not true. Present study design does not allow to do it.

In the introduction a nice overview is given of Histomonosis as a disease and briefly introduced previous risk factor studies. However, it is not clear weather similar descriptive studies have been performed previously elsewhere or what was the true motivation of the study. You report that this is the follow up study of a case-control study and you wanted to learn more about perceptions and awareness of farmers (lines 64-68). However, it is not clear from materials and methods or supplementary tables how did you investigate awareness and perceptions. From presented data it appears that you asked questions about what the farmers do, but it is not possible to make conclusions about perceptions and awareness solely based on behaviour of individuals. There is specific methodology how to investigate perceptions and awareness. You should not mention them as aims or results in this report unless you can provide evidence that you really have investigated these aspects.

In line 64 you state that "We implemented a case-control-study with 31 turkey farmers...". Please rephrase it as this is not a case-control study.

Material and Methods

In lines 73-85 you give an extensive description of the methodology of the preceding case-control study. Why it is necessary- are you reporting the results of the online questionnaire as well here. Please make this explicit and provide only necessary information regarding the C-C study.

In my understanding the description of the methods of the present study starts from line 86. 

Line 88: 'After a testing phase' Please give more detail, what do you mean - after testing of the questionnaire? On how many and whom? 

Line 92.93 '...he evaluation of the state of knowledge about histomonosis' It is not clear how this was evaluated and which data are representing this evaluation. Either remove or clarify.

Results and discussion

In general from case series studies we try to find and characterise common features of the cases (outbreak herds in these case). These common features give us ground to rise the hypothesis on risk factors or causes of the outbreaks. The results and discussions should be thoroughly reviewed in this respect and clearly more attention should be paid to factors that are similar to outbreak farms. At present these aspects disappear somewhat in the middle of the discussion around minor factors. There is too much discussion upon possible effects of factors being present only in few outbreak farms (Coincidental findings; e.g vaccination) and the conclusions or assumptions are unproportionally strong regarding their effects.

Line 208-210. Please rephrase without mentioning awareness. You investigated behaviour, and you can make conclusions about that.

Line 220-221: "Based on the analyses, it can be concluded that farmers are generally well aware of the risk of additional stress caused by vaccination [44]". I don't understand this. Based on analysis of what data? Where are they presented? Why there is a reference after this statement to a Chinese study? 

Conclusions

Need to be rewritten. Conclusion should be made based on study results. There cant be any conclusion about the effect of studied factors - only hypothesis about potential effects.

The clear distinction should be made between conclusions and recommendations.

The presented recommendations are also not related to study results and are based on other studies or assumptions, not evidence. This should be made explicit.

Please consider moving recommendation to the discussion part and provide references.

Some expressions should be checked with native speaker:

Line 176 'turkey house-specific overall'. Perhaps better 'turkey-house-specific overall', otherwise it reads 'turkey overall'

Line 266 'were controlled', Did you mean 'were inspected'?

Line 268 etc. 'are killed immediately' Why present perfect? In some sentences further in the past again. Please check.

Line 289 'could infest the contaminated dung'. How the contaminated dung could be infested?

Author Response

please, see the attachment

Reviewer 2 Report

In this manuscript, authors have presented findings of their field survey to identify possible factors associated with histomonosis outbreaks in German turkey flocks. 

In general, the manuscript is interesting and easy to understand. I have some minor comments: 

1. In line 68, authors mentioned that this study is a follow up of their previous work (Luenig et al., 2022). The time period of data collection in the previous and this study seems overlaping. Please make it clear in which aspect was this a follow up study? Probably, 31 participants in this study are among those 121 farmers from the previous one. Were they resurveyed? It was not clear to understand. 

2. Please provide  respective number of farms above each bar in Figure 2. 
